# Mixed Hepatocellular Cholangiocarcinoma: A Comparison of Survival between Mixed Tumors, Intrahepatic Cholangiocarcinoma and Hepatocellular Carcinoma from a Single Center

**DOI:** 10.3390/cancers15030639

**Published:** 2023-01-19

**Authors:** Lea Penzkofer, Lisa-Katharina Gröger, Maria Hoppe-Lotichius, Janine Baumgart, Stefan Heinrich, Jens Mittler, Tiemo S. Gerber, Beate K. Straub, Arndt Weinmann, Fabian Bartsch, Hauke Lang

**Affiliations:** 1Department of General, Visceral and Transplant Surgery, University Medical Center, Johannes Gutenberg-University Mainz, 55131 Mainz, Germany; 2Department of Pathology, University Medical Center, Johannes Gutenberg-University Mainz, 55131 Mainz, Germany; 31st Department of Internal Medicine, Gastroenterology and Hepatology, University Medical Center, Johannes Gutenberg-University Mainz, 55131 Mainz, Germany

**Keywords:** mixed hepatocellular cholangiocarcinoma, intrahepatic cholangiocarcinoma, hepatocellular carcinoma, liver surgery, tumor recurrence, overall survival

## Abstract

**Simple Summary:**

Mixed hepatocellular cholangiocarcinoma (mHC-CC) is a very rare tumor and data on its outcome after resection are scarce. The aim of this retrospective study was to compare recurrence and survival after surgery of mixed tumors with data from patients with pure hepatocellular (HCC) or intrahepatic cholangiocarcinoma (ICC). The most striking result was that mHC-CC showed a long-term outcome after resection comparable to ICC. Resection of non-cirrhotic HCC was associated with the longest survival, followed by HCC in cirrhosis. A small group of patients who underwent orthotopic liver transplant for mHC-CC had the best long-term outcome. The cholangiocarcinoma component of mHC-CC seems to be the defining outcome. Transplant within the Milan criteria might be a feasible option.

**Abstract:**

Background: Hepatocellular carcinoma (HCC) is the most frequent primary liver malignancy, followed by intrahepatic cholangiocarcinoma (ICC). In addition, there is a mixed form for which only limited data are available. The aim of this study was to compare recurrence and survival of the mixed form within the cohorts of patients with HCC and ICC from a single center. Methods: Between January 2008 and December 2020, all patients who underwent surgical exploration for ICC, HCC, or mixed hepatocellular cholangiocarcinoma (mHC-CC) were included in this retrospective analysis. The data were analyzed, focusing on preoperative and operative details, histological outcome, and tumor recurrence, as well as overall and recurrence-free survival. Results: A total of 673 surgical explorations were performed, resulting in 202 resections for ICC, 344 for HCC (225 non-cirrhotic HCC, ncHCC; 119 cirrhotic HCC, cHCC), and 14 for mHC-CC. In addition, six patients underwent orthotopic liver transplant (OLT) in the belief of dealing with HCC. In 107 patients, tumors were irresectable (resection rate of 84%). Except for the cHCC group, major or even extended liver resections were required. Vascular or visceral extensions were performed regularly. Overall survival (OS) was highly variable, with a median OS of 17.6 months for ICC, 26 months for mHC-CC, 31.8 months for cHCC, and 37.2 months for ncHCC. Tumor recurrence was common, with a rate of 45% for mHC-CC, 48.9% for ncHCC, 60.4% for ICC, and 67.2% for cHCC. The median recurrence-free survival was 7.3 months for ICC, 14.4 months for cHCC, 16 months for mHC-CC, and 17 months for ncHCC. The patients who underwent OLT for mHC-CC showed a median OS of 57.5 and RFS of 56.5 months. Conclusions: mHC-CC has a comparable course and outcome to ICC. The cholangiocarcinoma component seems to be the dominant one and, therefore, may be responsible for the prognosis. ‘Accidental’ liver transplant for mHC-CC within the Milan criteria offers a good long-term outcome. This might be an option in countries with no or minor organ shortage.

## 1. Introduction

Hepatocellular carcinoma (HCC) is the most frequent primary liver tumor, followed by intrahepatic cholangiocarcinoma (ICC). Together, they account for 85% of primary liver malignancies [1]. With HCC accounting for 70% and ICC for 12%, mixed hepatocellular cholangiocarcinoma (mHC-CC) is a rare combination of both tumors with a frequency of 2–3% [2]. The incidence of HCC as well as ICC is rising within the Western world, but ICC and especially mHC-CC still remain rare [3]. Mixed hepatocellular cholangiocarcinoma is a very heterogeneous tumor with differing amounts of hepatocellular or cholangiocarcinoma components and genomic profiles [4]. The 2019 WHO classification of tumors of the digestive system (5th edition, World Health Organization) addresses mixed tumors including mHC-CC, but openly communicates that this entity, among others, and its development as well as treatment remain subjects of uncertainty [5].

For HCC without cirrhosis (ncHCC, non-cirrhotic HCC) and ICC, surgical resection is the best therapeutic option that most likely offers the only chance of a cure [6,7,8,9]. Recent national and international guidelines or consensus statements support this approach [10,11,12,13]. HCC with cirrhosis (cHCC) must be considered separately as, in addition to resection, transplantation is also available as a therapy [14]. In European countries, the decision for or against resection or transplantation is made on the basis of the BCLC classification. [12,15]. Even for mHC-CC, surgical resection is the option of choice, while liver transplantation is also performed regularly [16,17]. Overall survival (OS) after surgical resection varies between the different entities: the 5-year OS range for ICC is between 20 and 40% [7,18,19,20,21], showing the worst outcome, while ncHCC shows 30–80% [22,23] and cHCC shows 50–61% [24,25]. For mHC-CC, the 5-year OS ranges between 28 and 63% [26,27]. Recent studies showed a worse long-term outcome after resection of mHC-CC in comparison with HCC, but comparable results with HCC and ICC after proper matching with regard to tumor burden [28,29]. Recurrence is frequent for all observed entities and defines long-term outcomes. While (palliative) systemic chemotherapy remains the most frequent treatment for recurrent ICC, patients with ncHCC as well as cHCC can benefit from different approaches, such as transarterial chemoembolization (TACE), chemotherapy, or less frequently selective internal radiation therapy (SIRT) [12].

The aim of this study was to compare the outcomes after resection of mHC-CC with large cohorts of patients with HCCwith or without cirrhosis and ICC from a single center. This study focused on overall survival and tumor recurrence of these different but overlapping entities. In addition, the long-term outcome of liver transplantation for mHC-CC within the Milan criteria was demonstrated.

## 2. Materials and Methods

This study was designed as a retrospective single-center analysis of the different entity cohorts listed below. The data from all patients who underwent surgical exploration for liver resection at our center were collected in our institutional database. A retrospective analysis was performed for all explorations or resections for intrahepatic cholangiocarcinoma (ICC), hepatocellular carcinoma (HCC), or mixed hepatocellular cholangiocarcinoma (mHC-CC) within the 13-year period from 2008 to 2020. Patients aged under 16 years or patients with primary liver tumors other than ICC, HCC, or mHC-CC were excluded from analysis.

Preoperative diagnostics included computer tomography (CT) or magnetic resonance tomography (MRI) of the abdomen as well as a CT scan of the thorax. Carbohydrate antigen 19-9 (CA19-9) value for ICC and alpha-fetoprotein (AFP) value for HCC were recorded as tumor markers. Liver function was assessed by means of bilirubin, Quick, and albumin values. We discussed all patients in an interdisciplinary tumor conference with experienced hepatobiliary surgeons, radiologists, and oncologists. Indications for resection were—irrespective of the tumor entity—the exclusion of distant metastasis, an absence of portal hypertension in preoperative imaging, and a non-compromised liver function in laboratory values, as well as the surgical-technical aspect of resectability. If the tumor appeared primary resectable, we waived percutaneous biopsy for diagnostic purposes. In the presence of HCC within the Milan criteria, a separate transplant board decided on the possibility of OLT in each individual case. Known ICC and mHC-CC were excluded from the possibility of OLT.

The surgery was performed by an experienced team of surgeons with special hepato-biliary expertise. For the classification of the resections, the “New World” terminology was used [30]. The postoperative follow-up was performed every three months for a minimum of two years after resection. At least every six months, we conducted CT or MRI in alternation of ultrasound examinations. The information was retrieved from the treating physicians if the patients were not able or not willing to undergo follow-up at our center. In principle, we excluded patients who underwent liver transplantation because it is only accessible for cHCC patients and our focus is on liver resection. We included six patients who underwent liver transplantation for mHC-CC. These patients were within the Milan criteria in the belief that the underlying disease was HCC in cirrhosis [31]. mHC-CC was diagnosed based on the final pathology of the explanted liver.

The data of the patients undergoing resection were further analyzed regarding preoperative treatment, performed resection with vascular or visceral extensions, pathological findings, and tumor recurrence and its primary treatment, as well as recurrence-free and overall survival. Morbidity was assessed according to the Dindo-Clavien classification [32]. For TNM staging, we used the 8th edition of the UICC classification (Union for International Cancer Control) [33].

An informed consent form has been signed by all patients that data and follow-up would be collected anonymously and potentially used for scientific analysis. This study is in accordance with the regulations of the federal state law (state hospital laws §36 and §37, Rhineland-Palatinate), and no ethical approval was necessary for this study according to the independent ethics committee of Rhineland-Palatinate,.

Statistical analyses were performed using SPSS 27 (IBM Corp. Released 2020. IBM SPSS Statistics for Windows, Version 27.0. Armonk, NY, USA: IBM Corp). For categorical data, Chi^2^ test was used in cross tabulation. The analyses of recurrence-free and overall survival were conducted using the Kaplan Meier model, and for comparison of influencing factors, a log rank test was utilized. Significance was considered with a *p*-value of < 0.05. Recurrence-free survival was defined according to Punt and colleagues [34].

## 3. Results

During the period of 2008 to 2020, a total of 673 patients underwent surgical exploration because of intrahepatic cholangiocarcinoma (n = 274), mixed hepatocellular cholangiocarcinoma (n = 14 plus additional 6 patients who underwent liver transplantation) or hepatocellular carcinoma (n = 379). A total of 107 tumors were irresectable due to varying reasons, leading to 202 resections for ICC, 14 (+6 transplants) for mHC-CC, and 369 for HCC (119 with and 225 without cirrhosis). See also Figure 1. Resectability was 73.7% for ICC and 90.8% for HCC.

### 3.1. Surgical Procedures and Intraoperative Data

A detailed overview of the resections performed can be found in Table 1. The proportions of extended, major, and minor resections differed significantly between the different entities (*p* < 0.001).

### 3.2. Vascular and Visceral Extensions

Table 1 shows the number of vascular and visceral extensions performed according to tumor entity. Overall, 21.8% of patients with ICC, 10% with mHC-CC, 14.3% with cHCC, and 19.3% with ncHCC underwent vascular resection. Resection of major hepatic vessels or inferior vena cava (IVC) was most common in ICC and ncHCC. Reconstruction of the hepatic artery was performed very rarely, with a total of four cases. Visceral extension of the primary resection was performed due to infiltration per continuitatem into the nearby organs. There was no difference in frequency according to tumor entity. Most commonly, parts of the diaphragm were resected followed by the right adrenal gland.

### 3.3. Histopathological Examination

Table 2 shows a summary of the pathological results. Preoperatively known distant metastasis was an exclusion criterion for resection. In addition, in individual cases, there was an intraoperative incidental finding of a localized distant tumor manifestation. Figure 2 shows the typical histopathological images of mixed hepatocellular cholangiocarcinoma.

### 3.4. Tumor Recurrence

Table 3 shows the data on frequency and location of recurrence stratified according to tumor entity.

For ICC, the most common therapy for recurrence was chemotherapy (n = 65, 32.2%), followed by resection (n = 6.9%) and best supportive care (BSC; n = 8, 4%). For mHC-CC, BSC was the therapy of choice in most patients (n = 5, 25%), followed by repeated resection (n = 2, 10%), TACE and chemotherapy (n = 1 each, 5%).

TACE was the most frequent therapy in patients treated for recurrence of cHCC (n = 30, 25.2%), followed by chemotherapy (n = 16, 13.4%) and repeated resection (n = 14, 11.8%). For ncHCC, the most common therapy for recurrence was chemotherapy (n = 34, 15.1%), followed by repeated resection (n = 26, 11.6%) and TACE (n = 23, 10.2%).

### 3.5. Comparison of Survival

A detailed evaluation can be found in the three survival curves. Both the OS and the RFS differ significantly between the tumor entities. For a better comparability between the resection groups, only patients with resection for mHC-CC were included. OS for mHC-CC after resection vs. OLT was considered separately.

#### 3.5.1. Overall Survival

Overall survival (OS) differs significantly according to tumor entity (see Figure 3). Patients with ICC had the shortest OS with a median of 17.6 months (range 0–132), followed by patients with mHC-CC (median OS of 26 months, range 0–48) and cHCC (median OS of 31.8 months, range 1–155). Patients with ncHCC showed the longest OS with a median of 37.2 months (range 1–156). The respective 1-, 3-, and 5-year OS are 73%, 34%, and 20% for ICC; 69%, 37%, and 37% for mHC-CC; 79%, 49%, and 28% for cHCC; and 77%, 60%, and 43% for ncHCC.

#### 3.5.2. mHC-CC Survival Resection vs. Transplant

The detailed information of the mHC-CC subgroup is shown in Table 4. The resection group showed a median overall survival of 26 months (range 0–48) and a consecutive 1-, 3-, and 5-year OS of 69%, 37%, and 37%. Regarding recurrence-free survival, the median RFS was 16 months (range 0–48).

The orthotopic transplantation group showed a median OS of 57.5 months (range 39–131) with a consecutive 1-, 3-, and 5-year OS of 100%, 100%, and 80%. The median RFS was 56.5 months (range 39–131).

#### 3.5.3. Recurrence-Free Survival

Recurrence-free survival (RFS) differs significantly according to tumor entity (see Figure 4). Similar to OS, patients with ICC had the shortest RFS with a median of 7.3 months (range 0–132), followed by patients with cHCC (median RFS of 14.4 months, range 1–130) and mHC-CC (median RFS of 16 months, range 0–48). Patients with ncHCC not only showed the longest OS but also the longest RFS with a median of 17 months (range 0–149). The respective 1-, 3-, and 5-year RFS are 47%, 26%, and 21% for ICC; 67%, 29%, and 29% for mHC-CC; 68%, 27%, and 21% for cHCC; and 68%, 50%, and 40% for ncHCC.

## 4. Discussion

Mixed hepatocellular cholangiocarcinoma is a rare entity and data on its therapy and outcome are scarce. This study offers insights using data from a single-center cohort with 560 resections for intrahepatic cholangiocarcinoma, hepatocellular carcinoma, or mixed hepatocellular cholangiocarcinoma, along with six orthotopic liver transplants in the latter group as well. The rarity of mHC-CC becomes apparent with an admittedly small subgroup of 20 patients. ncHCC showed the best overall survival (OS), followed by cHCC. mHC-CC had a comparable OS to ICC. Tumor recurrence was common in all entities but appeared least frequently in mHC-CC at a rate of 45%, and most frequently in cHCC at a rate of 67.2%. In a comparison between the mHC-CC resection and OLT groups, the OLT group showed a significantly better OS.

Beside its rarity, mHC-CC also shows a distinct heterogeneity [4]. Therefore, elaborate therapies and recommendations regarding every aspect of treatment, such as treatment in neoadjuvant, surgical, adjuvant, and palliative situations or in case of tumor recurrence, are not yet defined [16]. Complete resection with lymphadenectomy, at least in patients with ncHCC, is the therapy of choice if extrahepatic spread has been ruled out [35].

ICC and HCC have a clinical course causing no or minor symptoms for a long time. Therefore, both entities are often diagnosed in an advanced stage, making extended resections necessary for a chance of cure [36,37]. In our cohort, visceral and vascular extensions were performed regularly to achieve complete resection, as already reported in earlier studies [18,22]. Even within the mHC-CC group, four out of 14 patients (28.6%) underwent either visceral (n = 2) or vascular (n = 2) resections and reconstructions. For mixed tumors, there are hardly any data for comparison. For ICC and HCC, visceral and vascular extensions are commonly reported in the international literature [38,39,40,41].

In our cohort, the non-cirrhotic hepatocellular carcinoma group showed the best OS and RFS. With a different etiology in comparison to cirrhotic HCC, one must keep in mind that cirrhosis is a crucial life-limiting factor itself. This might be the reason for the worse outcome. The 5-year OS for the ncHCC and cHCC groups were 43% and 28%, respectively. In the literature, the 5-year OS varies widely between 30 and 80% for ncHCC [23] and between 42 and 55% for cHCC [23,42,43]. Our results are in the lower range. This might be due to the large number of major liver resections. Besides, visceral or vascular extensions were performed outstandingly often. The survival of the ICC group was poorest, with a 5-year OS of 20%, out of the four included entities. This is within the range of recent publications, which report a range from 20 to 45% [7,21,44,45]. Even for the ICC cohort, the argument of a vast number of major resections and extensions applies as well.

Data on mHC-CC are scarce and, even in the present subgroup of patients who underwent resection, the number is small with only 14 patients. The median OS was 26 months with an estimated 5-year OS of 37%. More recently, Leoni and colleagues published a comprehensive review of the literature with a median OS ranging between 18.3 and 52.5 months. Comparable but extremely varying results showed a 5-year OS ranging from 10.5 to 66 months [16]. It must be considered that, in many studies, the results after liver transplant were included. The 5-year OS for resection only ranges from 36.4% to 63% in cohorts with 68, 100, and 103 patients who underwent resection [27,46,47].

Recent studies addressed mixed hepatocellular cholangiocarcinoma and compared it with intrahepatic cholangiocarcinoma and/or hepatocellular carcinoma. In a systematic review, Gentile and colleagues compared long-term outcome after resection or transplantation of mHC-CC with HCC. Regarding both overall and disease-free survival, mHC-CC performed significantly worse [28]. A detailed review by Beaufrère and colleagues comes to the conclusion that the prognosis of mHC-CC is similar to ICC, but worse than HCC [48]. The findings of both articles are in accordance with our results. In another publication by Gentile and colleagues, the outcome of mHC-CC was compared with HCC and mass-forming ICC in a case-matched analysis. After matching for tumor burden, all entities showed comparable overall survival [29].

Tumor recurrence is a major therapeutic challenge and defines the course of the disease after the initial complete resection. With frequencies ranging between 45% and 67.2% in the four studied groups, recurrence was common. Isolated intrahepatic recurrence as the first manifestation was by far the most frequent. Except for the cHCC group where almost all tumors recurred only intrahepatically, isolated extrahepatic or combined intra- and extrahepatic recurrence occurred regularly as well. For ICC, the frequency of tumor recurrence is reported to range between 61% and 73.4% [49,50,51,52], which is in accordance with our finding of 60.4%. Due to some patients having incomplete follow-up information, the percentage might be even slightly higher. Approaches with curative intention are scarce and limited to repeated resection or ablation in highly selected patients [53,54,55,56]. In most cases, systemic chemotherapy is recommended and applied [10,57,58]. For cHCC and ncHCC, the recurrence rates in our study were 67.2% and 48.9%, respectively. This is also in accordance with the literature, with rates ranging from 54% to 66% [59,60,61]. In contrast to ICC, there exists a variety of different treatment approaches for HCC, offering higher effectiveness with curative, time-gaining, or palliative intentions. Repeated resection is an alternative, along with ablation, transarterial chemoembolization (TACE), selective internal radiation therapy (SIRT), or chemotherapy [13,62,63,64]. A study by Erridge and colleagues compared survival following repeated resection, ablation, and TACE in recurrent HCC and found no significant differences in long-term outcome [64]. For mHC-CC, tumor recurrence is also common and ranges somewhere between 42% and 86.6% [16,27,65]. Due to the small number of reported cases, recommendations regarding therapy of recurrence are understandably rare, but repeated resection, TACE, ablation, and chemotherapy are possible therapeutic options [66].

As we placed our focus on curative intended resection, we did not include patients who underwent OLT, especially for the cHCC group. We refer to the large number of publications that address OLT for HCC with cirrhosis. Even for ICC, OLT was performed accidentally in a few patients at our center with poor long-term outcome. These patients are likewise not reported. Nevertheless, we wanted at least to mention and analyze the outcome of mHC-CC patients who underwent OLT in the belief of dealing with HCC. We found good overall and recurrence-free survival with medians of 57.5 and 56.5 months, respectively. It is important to mention that the OLT tumors were within the Milan criteria [31]. Due to the more aggressive cholangiocarcinoma component of mHC-CC, advanced tumors might not be good candidates for liver transplant. Our estimated 5-year OS was 80%, which is superior to the reported rates of 16%, 39%, 50%, and 66% for OLT patients with mHC-CC [27,46,67,68]. Most likely, the poor outcomes can be explained by the inclusion of patients with advanced disease.

This study has some limitations. The retrospective design might lead to reduced validity. Furthermore, the subgroup of patients with mixed hepatocellular cholangiocarcinoma is small which might lead to a relevant bias. A higher number of patients would have been crucial, but with high numbers of resection for ICC and HCC, the rareness of mHC-CC becomes apparent.

## 5. Conclusions

Resection of hepatocellular carcinoma is associated with a better long-term outcome than mixed hepatocellular cholangiocarcinoma or intrahepatic cholangiocarcinoma. Although cHCC has the highest recurrence rate after resection, it still surpasses ICC and mHC-CC in terms of overall survival. This is mainly due to the more effective therapeutic options in case of recurrence. The cholangiocarcinoma component of mHC-CC seems to have prognostic relevance, leading to the moderate OS.

The long-term outcome and recurrence-free survival after orthotopic liver transplant for mHC-CC within the Milan criteria are good. Especially in countries or health care systems with minor organ shortage, even transplantation seems to be a reasonable approach, given the low rate of tumor recurrence.

## Figures and Tables

**Figure 1 cancers-15-00639-f001:**
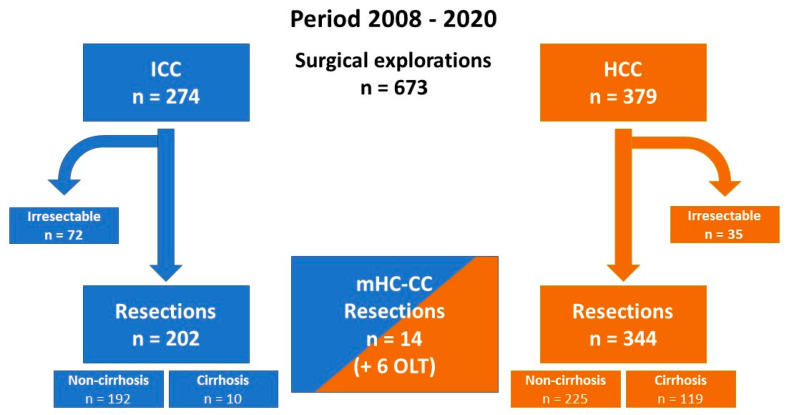
Flow chart of explorations and resection for intrahepatic cholangiocarcinoma (ICC), hepatocellular carcinoma (HCC), and mixed hepatocellular cholangiocarcinoma (mHC-CC). OLT = orthotopic liver transplantation.

**Figure 2 cancers-15-00639-f002:**
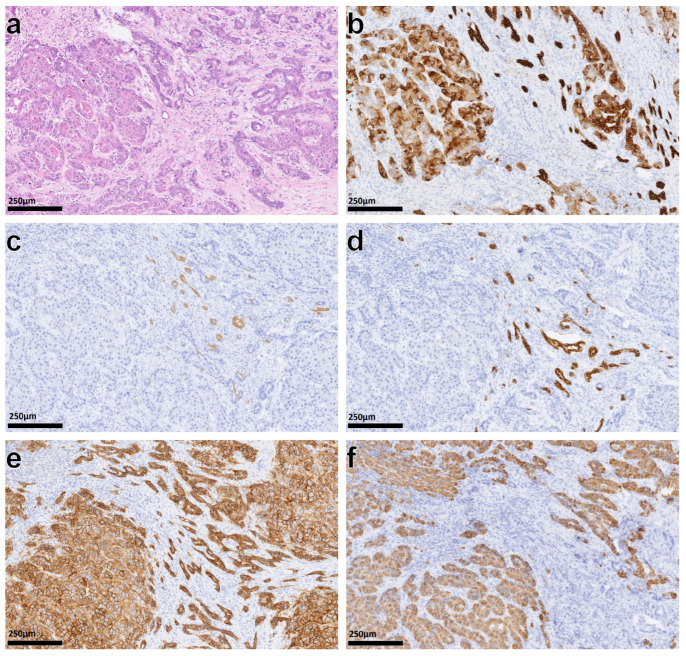
Histopathological image of mixed hepatocellular cholangiocarcinoma (**a**) with unequivocal hepatocytic (bottom left) and cholangiocytic (top right) areas of differentiation. Immunohistochemistry demonstrates the biphenotypic nature of the tumor. The tumor shows positivity for cholangiocytic markers ((**b**), CK 7) and focally prominent markers of stem cell differentiation ((**c**), CD56 and (**d**), CK19). Usually absent in hepatocellular carcinomas, this tumor expresses BerEp4 (**e**) and, in most areas, Hepar1, a marker of hepatocytic lineage (**f**).

**Figure 3 cancers-15-00639-f003:**
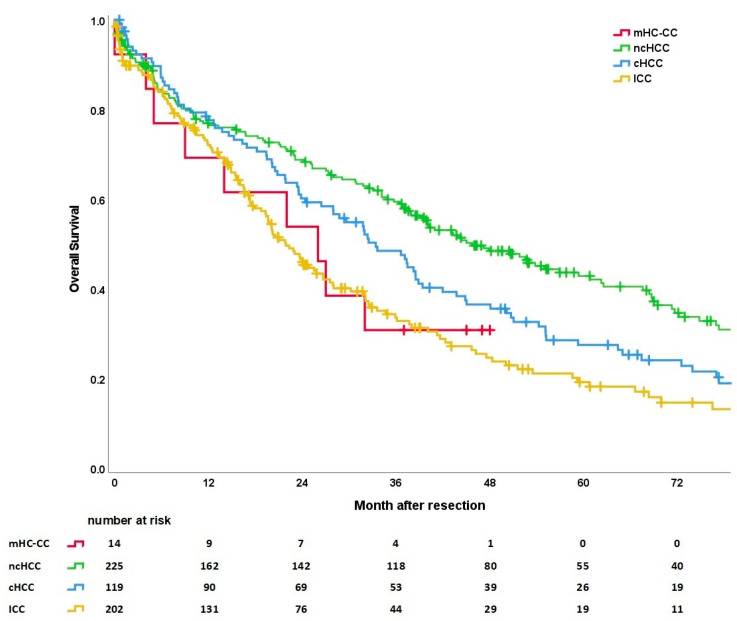
Overall survival curves according to tumor entity (*p* < 0.001). Subgroups mHC-CC vs. ncHCC (*p* = 0.089), mHC-CC vs. cHCC (*p* = 0.390), mHC-CC vs. ICC (*p* = 0.882), ncHCC vs. cHCC (*p* = 0.029), ncHCC vs. ICC (*p* < 0.001), and cHCC vs. ICC (*p* = 0.033). Six patients with OLT for mHC-CC were excluded.

**Figure 4 cancers-15-00639-f004:**
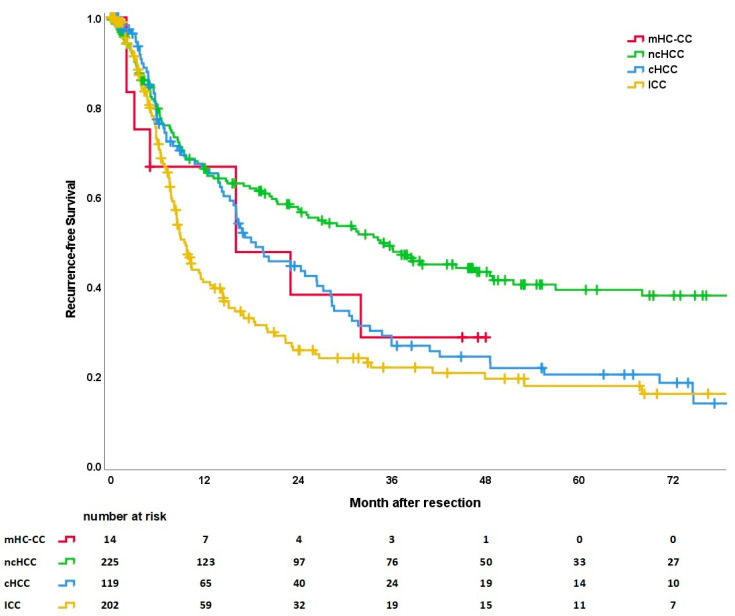
Recurrence-free survival curves according to tumor entity (*p* < 0.001). Subgroups mHC-CC vs. ncHCC (*p* = 0.244), mHC-CC vs. cHCC (*p* = 0.926), mHC-CC vs. ICC (*p* = 0.479), ncHCC vs. cHCC (*p* = 0.003), ncHCC vs. ICC (*p* < 0.001), and cHCC vs. ICC (*p* = 0.040). Six patients with OLT for mHC-CC were excluded.

**Table 1 cancers-15-00639-t001:** Baseline characteristics and surgical data.

	ICC	mHC-CC	cHCC	ncHCC
	n = 202	n = 20	n = 119	n = 225
Gender (Female/Male)	98/104	5/15	24/95	50/175
Age [Median (IQR)]	64.4 (57.4–74)	62.5 (56–67)	69.9 (63.7–75)	70.8 (63–76.1)
ASA classification				
ASA I	2	0	0	1
ASA II	98	7	43	57
ASA III	105	13	73	158
ASA IV	6	0	3	9
Resections				
Extended resections	82 (40.6%)	6 (30%)	5 (4.2%)	42 (18.6%)
Right trisectionectomy	32	3	2	25
Left trisectionecotmy	32	1	1	8
Mesohepatectomy	11	1	2	9
ALPPS	7	1	0	0
Major resections	71 (35.1%)	2 (10%)	19 (16%)	85 (37.8%)
Right hepatectomy	27	2	10	51
Left hepatectomy	31	0	5	20
Trisegmentectomy	13	0	4	13
Bisectionectomy	0	0	0	1
Minor resections	49 (24.3%)	6 (30%)	95 (79.8%)	98 (43.6%)
Bisegmentectomy	33	4	33	47
Monosegmentectomy	14	1	37	29
Atypical/wedge resections	2	1	25	22
Liver transplantation	-	6 (30%)	-	-
Vascular resection/reconstruction *	67 in 47 pat.	2 in 2 pat.	20 in 17 pat.	52 in 45 pat.
PV/MHV/HA/IVC	26/22/2/17	1/1/0/0	10/7/1/2	14/21/1/16
Visceral resection/reconstruction *	21 in 19 pat.	2 in 2 pat.	12 in 11 pat.	28 in 27 pat.
Adrenal gland/Diaphragm/Duodenum/Stomach/Colon/Pericardium	5/12/1/1 /1/1	1/1/0/0 /0/0	3/5/0/3 /1/0	7/18/1/0 /2/0
Number of lesions				
n = 1/2/3/ multifocal (≥4)	150/14/11/27	14/1/0/5	72/15/8/24	162/24/8/31
Lymphadenectomy [number performed (%)]	181 (89.6%)	14 (70%)	37 (31.1%)	98 (43.6%)
Lymph nodes harvested [Median (IQR)]	5 (2–8)	1.5 (0–3)	2 (1–3)	2 (1–5)

ICC: intrahepatic cholangiocarcinoma; mHC-CC: mixed hepatocellular-cholangiocellular carcinoma; cHCC: hepatocellular carcinoma with cirrhosis; ncHCC: hepatocellular carcinoma with non-cirrhosis; IQR: interquartile range; ASA: American Society of Anaesthesiologists; pat.: patients; PV: portal vein; MHV: major hepatic vein; HA: hepatic artery; IVC: inferior vena cava; * in some patients, multiple resections/reconstructions were performed; therefore, the number of interventions might differ from the number of patients.

**Table 2 cancers-15-00639-t002:** Histological outcomes after resection.

	ICC	mHC-CC	cHCC	ncHCC
	n = 202	n = 20	n = 119	n = 225
Resection, n (%)				
R0	166 (82.2)	18 (90)	99 (83.2)	198 (88.0)
R1	35 (17.3)	2 (10)	12 (10.1)	16 (7.1)
R2	1 (0.5)	0	7 (5.9)	5 (2.2)
Rx	0	0	1 (0.8)	6 (2.7)
T stage, n (%)				
T1 (a/b)	85 (34/51; 42.1)	11 (55)	53 (44.5)	79 (35.1)
T2	76 (37.6)	6 (30)	29 (24.4)	76 (33.8)
T3	15 (7.4)	3 (15)	30 (25.2)	54 (24.0)
T4	26 (12.9)	0	6 (5.0)	15 (6.7)
N category, n (%)				
N0	123 (60.9)	12 (60)	36 (30.3)	92 (40.9)
N1	58 (28.7)	2 (10)	1 (0.8)	6 (2.7)
Nx	21 (10.4)	6 (30)	82 (68.9)	127 (56.4)
M category, n (%)				
M0	188 (93.1)	20 (100)	117 (98.3)	212 (94.2)
M1	14 (6.9)	0	2 (1.7)	13 (5.7)
Tumour Grading, n (%) *				
G1	3 (1.5)	1 (5)	11 (9.2)	15 (6.7)
G2	129 (63.9)	8 (40)	75 (63.0)	132 (58.7)
G3	51 (25.2)	7 (35)	31 (26.1)	69 (30.7)
G4	1 (0.5)	0	0	6 (2.7)
Vascular Invasion, n (%)				
V0	155 (76.7)	12 (60)	76 (63.9)	112 (49.8)
V1	43 (21.3)	7 (35)	31 (26.1)	87 (38.7)
V2	4 (2)	1 (5)	12 (10.1)	25 (11.1)
Largest nodule diameter (mm), [median, range]	67 (4–200)	48 (8–130)	47 (12–160)	85 (10–300)

* Patients who underwent neoadjuvant treatment had no grading after resection.

**Table 3 cancers-15-00639-t003:** Location of tumor recurrence.

	ICC	mHC-CC	cHCC	ncHCC
	n = 202	n = 14/6 OLT	n = 119	n = 225
Recurrence, n (%)	122 (60.4%)	8/1 (45%)	80 (67.2%)	110 (48.9%)
Intrahepatic Rec. (n)	52 (42.6%)	4/0 (44.4%)	66 (82.5%)	66 (60%)
Extrahepatic Rec. (n)	28 (23%)	1/0 (11.1%)	4 (5%)	15 (13.6%)
Intra- + extrahepatic Rec. (n)	42 (34.4%)	3/1 (44.4%)	10 (12.5%)	29 (26.4%)

ICC: intrahepatic cholangiocarcinoma; mHC-CC: mixed hepatocellular-cholangiocellular carcinoma; cHCC: hepatocellular carcinoma with cirrhosis; ncHCC: hepatocellular carcinoma with non-cirrhosis; rec: recurrence; OLT: orthotopic liver transplant.

**Table 4 cancers-15-00639-t004:** mHC-CC patients with surgical procedures, histology, tumor recurrence, and survival.

Pat. #	OP Year	Resection Type	Age	TNM Classification	Cirrhosis/Fibrosis	TTR	Rec. Localization	Rec. Therapy	OS	Status
Resection										
1	2011	Right trisectionectomy	64	T3, N1 (2/2), L1, V1, Pn0, G3, R0	none	-	-	-	5	dead
2	2012	Atypical resection	63	T1, Nx, L0, V0, Pn0, G2, R0	cirrhosis	16	liver/bone	BSC	22	dead
3	2013	Bisegmentectomy	62	T1, N1 (9/10), L0, V0, Pn1, G3, R0	septal fibrosis +ad				-	LFU
4	2014	Mesohepatectomy	72	T3, N0 (0/3), L0, V2, Pn0, G3, R1	cirrhosis	16	liver	resection	32	dead
5	2015	Bisegmentectomy	75	T3, N0 (0/1), L0, V0, Pn0, G3, R0	septal fibrosis -ad	5	liver, adrenal gl.	BSC	9	dead
6	2016	Left Trisectionectomy	68	T2, N0 (0/4), L0, V1, Pn0, G3, R0	portal fibrosis	2	liver	chemo	14	dead
7	2016	Left Hepatectomy	81	T2, N0 (0/2), L0, V1, Pn0, G2, R0	portal fibrosis	23	liver, kidney, adrenal gl.	BSC	26	dead
8	2016	Right trisectionectomy	48	T1, Nx, L0, V0, Pn0, Gx*, R0	cirrhosis	2	lung, brain	BSC	4	dead
9	2017	ALPPS	23	T2, N0 (0/14), L1, V1, Pn0, G2, R1	portal fibrosis	-	-	-	0	dead
10	2018	Right Trisectionectomy	67	T2, N0 (0/6), L0, V1, Pn0, G2, R0	septal fibrosis -ad	32	liver	TACE	37	alive
11	2018	Monosegmentectomy	67	T1, Nx, L0, V0, Pn0, G2, R0	cirrhosis	-	-	-	48	alive
12	2018	Right Hepatectomy	64	T1, N0 (0/3), L0, V0, Pn1, G3, R0	septal fibrosis +ad	-	-	-	45	alive
13	2018	Bisegmentectomy	61	T1, Nx, L0, V0, Pn0, G3, R0	septal fibrosis +ad	-	-	-	47	alive
14	2019	Bisegmentectomy	45	T1, N0 (0/2), L0, V1, Pn0, G2, R0	none	3	liver	Chemo	27	alive
Transplant										
15	2011	oLT	37	T1, N0 (0/2), L0, V0, Pn0, G2, R0	cirrhosis	-	-	-	131	alive
16	2013	oLT	59	T2, N0 (0/1), L0, V0, Pn0, G1, R0	cirrhosis	-	-	-	110	alive
17	2016	oLT	56	T2, Nx, L0, V1, Pn0, G2, R0	cirrhosis	44	liver	radiation	57	dead
18	2016	oLT	61	T1, N0 (0/1), L0, V0, Pn0, G2, R0	cirrhosis	-	-	-	39	dead
19	2017	oLT	64	T1, Nx, L0, V0, Pn0, G2, R0	cirrhosis	-	-	-	58	alive
20	2018	oLT	56	T1, N0 (0/1), L0, V0, Pn0, G1, R0	cirrhosis	-	-	-	55	alive

Pat. # = Patient number; OP year = year of operation; +ad = with architectural distortion; -ad = without architectural distortion; TTR = time to recurrence; Rec. = recurrence; adrenal gl. = adrenal gland; OS = overall survival; LFU = lost to follow-up; oLT = orthotopic liver transplant; * patient with no grading after neoadjuvant treatment.

## Data Availability

The datasets used and analyzed during the current study are available from the corresponding author upon reasonable request.

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
