# Peer review of "Mixed Hepatocellular Cholangiocarcinoma: A Comparison of Survival between Mixed Tumors, Intrahepatic Cholangiocarcinoma and Hepatocellular Carcinoma from a Single Center"

_cancers, 2023, doi:10.3390/cancers15030639_

Round 1

Reviewer 1 Report

Dear authors, I have read with interest this manuscript that concerns the interesting topic of mixed HCC-iCCA cases undergoing hepatic resection. The paper is of interest, but some changes should be provided:

1. Please provide english proofreading, in particular for the abstract that is not easy to read. Please consider that too much abbreviations may complicate the reading.

2. Please add more details in methods starting with study design.

3. Indications for resection for HCC, iCCA, and mHCC as well as for OLT should be added.

4. The paper is not overall very original, since some other similar literature have been found (i.e., 1: Gentile et al. Is the outcome after hepatectomy for transitional hepatocholangiocarcinoma different from that of hepatocellular carcinoma and mass-forming cholangiocarcinoma? A case-matched analysis. Updates Surg. 2020 Sep;72(3):671-679. doi: 10.1007/s13304-020-00802-w. Epub 2020 May 22. PMID: 32445033. || Gentile D et al. Surgical Treatment of Hepatocholangiocarcinoma: A Systematic Review. Liver Cancer. 2020 Jan;9(1):15-27. doi: 10.1159/000503719. Epub 2019 Nov 1. PMID: 32071906; PMCID: PMC7024854.) These papers should be added.

5. The comparison between resection and OLT is unfair either for the different cases, small sample size and retrospective nature of the comparison that includes many biases. I do not think that this comparison is an added value in this paper. 

Author Response

Comments for the Reviewers and Editors

We want to thank the reviewers for their time and effort for reviewing our manuscript. Changes are marked yellow within the manuscript.

Reviewer 1

Dear authors, I have read with interest this manuscript that concerns the interesting topic of mixed HCC-iCCA cases undergoing hepatic resection. The paper is of interest, but some changes should be provided:

  1. Please provide english proofreading, in particular for the abstract that is not easy to read. Please consider that too much abbreviations may complicate the reading.

We have critically reviewed the paper again for linguistic deficiencies and corrected them accordingly. In particular, the abstract has been smoothed out.

Regarding the abbreviations, we can absolutely understand your input. It is a bit of a challenge, because many reviewers demand that abbreviations should always be used. We addressed your comment and reduced the use of abbreviations.

  1. Please add more details in methods starting with study design.

Thank you very much for this helpful comment. We added the necessary details concerning study design as well as other relevant information in the method section.

  1. Indications for resection for HCC, iCCA, and mHCC as well as for OLT should be added.

A very helpful hint, we have included the corresponding information in the method section.

  1. The paper is not overall very original, since some other similar literature have been found (i.e., 1: Gentile et al. Is the outcome after hepatectomy for transitional hepatocholangiocarcinoma different from that of hepatocellular carcinoma and mass-forming cholangiocarcinoma? A case-matched analysis. Updates Surg. 2020 Sep;72(3):671-679. doi: 10.1007/s13304-020-00802-w. Epub 2020 May 22. PMID: 32445033. || Gentile D et al. Surgical Treatment of Hepatocholangiocarcinoma: A Systematic Review. Liver Cancer. 2020 Jan;9(1):15-27. doi: 10.1159/000503719. Epub 2019 Nov 1. PMID: 32071906; PMCID: PMC7024854.) These papers should be added.

We are sorry that we did not include the mentioned studies. Maybe the different term of ‘hepatocholangiocarcinoma’ may explain that we did not found them within our Pubmed-searches. We included them within the introduction and addressed both extensively within the discussion.

  1. The comparison between resection and OLT is unfair either for the different cases, small sample size and retrospective nature of the comparison that includes many biases. I do not think that this comparison is an added value in this paper.

You are right. The comparison underlies different reasons for bias limiting the validity. Especially the diverse preconditions of both groups. We removed the comparison, the regarding figure and changed the subsection within the results accordingly. Further, we adapted also the abstract and discussion.

Thank you again for your comments and help to improve our manuscript.

Lea Penzkofer, Hauke Lang and Fabian Bartsch

Reviewer 2 Report

Although this is a very interesting article regarding an update of mixed HCC-Choalangiocarcinoma, MINOR revisions could improve article's quality.

1. In the Introduction section, the number of paragraphs should be reduced. Additional data regading this rare entity should be included. The aim of the study should be more clear.

2. Parallel evaluation with recent studies' evidence should be provided in the Discussion section.

3. Tables are absolutely acceptable.

4. Newly published articles should be included in the References section.

5. Grammatical errors should be corrected throughout the Text.

4. 

Author Response

Comments for the Reviewers and Editors

We want to thank the reviewers for their time and effort for reviewing our manuscript. Changes are marked yellow within the manuscript.

Reviewer 2

Although this is a very interesting article regarding an update of mixed HCC-Choalangiocarcinoma, MINOR revisions could improve article's quality.

  1. In the Introduction section, the number of paragraphs should be reduced. Additional data regading this rare entity should be included. The aim of the study should be more clear.

We reduced the number of paragraphs and added the additional data accordingly. Further we extended the description of the studies aim.

  1. Parallel evaluation with recent studies' evidence should be provided in the Discussion section.

We added a new paragraph within the discussion section to address your comment.

  1. Tables are absolutely acceptable.

Thank you. Nothing to respond.

  1. Newly published articles should be included in the References section.

Thank you for your critical comment. We have reviewed the current literature and updated the discussion and the bibliography accordingly. See also comment #2.

  1. Grammatical errors should be corrected throughout the Text.

Thank you for your critical comment. We have reviewed the entire manuscript again for linguistic deficiencies and corrected them accordingly.

Thank you again for your comments and help to improve our manuscript.

Lea Penzkofer, Hauke Lang and Fabian Bartsch